# Noncanonical mechanism of Nrf2 activation by diacylglycerol polyethylene glycol adducts in normal human epidermal keratinocytes

**Tatsuro Miyoshi**[1]*, **Brian C. Keller**[1], **Takashi Ashino**[2], **Satoshi Numazawa**[2]

**1** Beverly Glen Laboratories, Inc. Newport Beach, Newport Beach, CA, United States of America,
**2** Department of Pharmacology, Division of Toxicology, Toxicology and Therapeutics, Showa University School of Pharmacy, Shinagawa, Tokyo, Japan

* tmiyoshi@bglen.jp

**Data Availability Statement:** All relevant data are within the paper and its Supporting Information files.

## Abstract

Polyethylene glycol-23 glyceryl distearate (GDS-23), a diacylglycerol polyethylene glycol adduct, forms niosomes with a liposome-like structure and functions as an active ingredient in drug delivery systems. In addition, it upregulates antioxidant proteins such as heme oxygenase 1 and NAD(P)H-quinone dehydrogenase 1 in cells. However, the activation of nuclear factor E2-related factor-2 (Nrf2), which plays a role in inducing the expression of antioxidant proteins, and its protective effects induced by GDS-23 treatment against oxidative stress have not been elucidated. This study aimed at verifying the activation of Nrf2 by GDS-23 and clarifying its underlying mechanisms, and investigated whether GDS-23 protects against hydroquinone-induced cytotoxicity. Normal human epidermal keratinocytes were treated with GDS-23. Real-time reverse transcription-polymerase chain reaction, western blotting, and immunostaining were used to investigate the mechanism of Nrf2 activation, and neutral red assay was performed to evaluate cytotoxicity. GDS-23-treated cells showed an increase in antioxidant protein levels and stabilization of Nrf2 in the nucleus. During Nrf2 activation, p62, an autophagy-related adaptor protein, was phosphorylated at Ser349. Inhibition of the interaction between the phosphorylated p62 and Kelch-like ECH-associated protein 1 significantly suppressed the GDS-23-mediated induction of antioxidant protein expression. In addition, hydroquinone-induced cell toxicity was significantly attenuated by GDS-23. GDS-23 induced the intracellular antioxidant system by activating Nrf2 in a p62 phosphorylation-dependent manner without generating oxidative stress in the cells. GDS-23 may be applied as a multifunctional material for drug delivery system that enhances internal antioxidant systems.

## Introduction

Niosomes, which are nano-sized vesicles comprising nonionic surfactants, have a liposome-like structure made of phospholipids. They are easier to prepare, less expensive, and more stable than liposomes [1]. Niosomes, which have multiple layers of hydrophilic and hydrophobic groups, can contain water-soluble and oil-soluble drugs, and are being investigated as carriers

**Funding:** This study was funded by Beverly Glen Laboratories, Inc.https://www.bglen.com TM is an employee of Beverly Glen Laboratories, Inc. BCK is Chief Science Officer of Beverly Glen Laboratories, Inc. TM conceived the idea for this study, conducted the experiments, and wrote the paper. BCK contributed to analysis of the results. The funders had no role in study design, data collection and analysis, decision to publish, or preparation of the manuscript.

**Competing interests:** I have read the journal's policy and the authors of this manuscript have the following competing interests. T. Miyoshi is an employee of Beverly Glen Laboratories, Inc. Brian C. Keller is Chief Science Officer of Beverly Glen Laboratories, Inc. S. Numazawa and T. Ashino serves as a member of a joint research project between Beverly Glen Laboratories, Inc. and Showa University. The authors declare that they have no known competing financial interests or personal relationships that could have appeared to influence the work reported in this paper. This does not alter our adherence to PLOS ONE policies on sharing data and materials.

PEG-23 Glyceryl Distearate (GDS-23)

**Fig 1. Typical structural formula of diacylglycerol polyethylene glycol adduct.**

in drug delivery systems (DDS) together with liposomes [2–5]. We have previously developed a niosome-based DDS using diacylglycerol polyethylene glycol (PEG) adduct, a nonionic surfactant (Fig 1). We found that the diacylglycerol PEG adduct easily forms multi-layered niosomes and enhances the skin penetration of drugs [6–8]. Furthermore, the diacylglycerol PEG adduct not only functions as a drug transport system but also inhibits cellular phospholipase A and cyclooxygenase-2 (COX2) expression and exerts anti-inflammatory effects [9]. Additionally, the diacylglycerol PEG adduct exerts a moisturizing effect by promoting the production of filaggrin in the epidermis [10] and an antioxidant effect by upregulating the expression of antioxidant-related genes in the epidermis [11]. These findings indicate that niosomes formed from diacylglycerol PEG adduct have multiple functions. Our preliminary studies suggested that the mechanism underlying the anti-inflammatory and antioxidant effects of diacylglycerol PEG adducts involves their action on the nuclear factor E2-related factor-2 (Nrf2)-Kelch-like ECH-associated protein 1 (Keap1) signaling pathway that regulates the intracellular antioxidant system [12]. Therefore, we hypothesized that the diacylglycerol PEG adduct may modulate the antioxidant system in the body and exert a protective effect against oxidative stress caused by air pollutants and ultraviolet (UV) rays [13], which have recently been associated with various skin problems.

The skin is the outermost layer of the human body and is an important tissue that protects various organs in the body from external stimuli and foreign substances; it maintains homeostasis of biological functions. Oxidative stress in the skin is a major factor that affects the aging process [14]. Oxidative stress caused by UV exposure increases COX2 expression and prostaglandin E2 synthesis [15].

Homeostasis of biological functions is maintained via the intracellular antioxidant system, which is a defense mechanism against oxidative stress. Several antioxidant enzymes and proteins alleviate oxidative stress, which is caused by the generation of excess intracellular reactive oxygen species (ROS). The intracellular antioxidant system is regulated by Nrf2-Keap1 signaling [16]. Under normal conditions, Nrf2 is localized to the cytoplasm in the form of a complex with Keap1 and is polyubiquitinated and degraded by the Keap1-dependent ubiquitin-proteasome system. Oxidative stress or exposure to electrophiles induces the oxidation of redox-sensitive cysteine residues of Keap1, and Nrf2 is released via the Keap1-dependent degradation mechanism and translocated to the nucleus. Subsequently, Nrf2 binds to antioxidant response elements as a heterodimer containing a small Maf protein, thereby enhancing the gene expression of antioxidant proteins [17]. Recently, it has been reported that the phosphorylation of p62, an autophagy-related adaptor protein [18], at Ser349 increases its binding to Keap1 and stabilizes Nrf2 by releasing it from Keap1 [19]. Other studies have reported that peroxisome proliferator receptor γ (PPARγ) is involved in the oxidative stress response and interacts with Nrf2 signaling in defense responses [20,21]. In this study, we investigated the effects of

diacylglycerol PEG adducts, which inhibit COX2 expression and upregulate intracellular antioxidant gene expression, on the Nrf2-Keap1 signaling pathway. We focused on the phosphorylation of p62 as the mechanism underlying the induction of nuclear translocation of Nrf2 by diacylglycerol PEG adduct. In addition, we investigated whether the antioxidant effect of diacylglycerol PEG adduct could reduce the cytotoxic effects of hydroquinone, which could produce ROS in the skin.

## Materials and methods

### Reagents

Polyethylene glycol-23 glyceryl distearate (GDS-23, Fig 1) was obtained from J-Network, Inc. (Newport Beach, CA, USA). ReverTra Ace qPCR RT Master Mix was purchased from Toyobo Co. (Osaka, Japan). PowerUp SYBR Green Master Mix was procured from Thermo Fisher Scientific (Waltham, MA, USA). BCA Protein Assay Kit was purchased from Takara Bio Inc. (Shiga, Japan). Alexa Fluor 488 goat anti-rabbit IgG (H+L) was purchased from Thermo Fisher Scientific. Cellstain-4′,6-diamidino-2-phenylindole (DAPI) solution was obtained from Dojindo Laboratories Co. (Kumamoto, Japan). K67 (N,N'-(2-(2-oxopropyl)naphthalene-1,4-diyl)bis(4-ethoxybenzenesulfonamide), N-[2-acetonyl-4-(4-ethoxybenzenesulfonylamino) naphthalene-1-yl]-4-thoxybenzenesulfonamide, and 2-acetonyl-1,4-bis[(4-2-acetonyl-1,4-bis [(4-thoxybenzensulfonyl)amino]naphthalene) were purchased from Sigma–Aldrich (St. Louis, MO, USA). Hydroquinone was purchased from Fujifilm Wako Pure Chemicals Corporation (Osaka, Japan). Neutral red solution (0.33%) was procured from Sigma–Aldrich. Polymerase chain reaction (PCR) primer sets were bought from Takara Bio; their sequences are listed in Table 1.

### Cell culture

Normal human epidermal keratinocytes (NHEKs) were purchased from Kurabo (Tokyo, Japan). The cells were cultured in HuMedia-KG2 (Kurabo) medium at 37°C under 5% $CO_2$. Subsequently, the cells were detached using 0.25% trypsin and 0.01% EDTA and then reseeded on plates and cultured for the following experiments.

### Cell viability

NHEKs were seeded into a 96-well plate at a density of $2.0 \times 10^4$ cells/well and cultured for 24 h using either HuMedia-KB2 containing GDS-23 or HuMedia-KB2 alone. In some experiments, NHEKs were pre-treated with GDS-23 for 24 h and cultured for an additional 24 h in HuMedia-KB2 medium containing 200 μM hydroquinone. NHEKs were also treated with 200 μM hydroquinone in the presence of 500 μM sodium pyrosulfite for 24 h. After treatment

**Table 1. Primers used in quantitative polymerase chain reaction.**

| Gene | Forward primer (5'-3') | Reverse primer (5'-3') |
|---|---|---|
| *Glyceraldehyde-3-phosphate dehydrogenase* (*GAPDH*) | GCACCGTCAAGGCTGAGAAC | TGGTGAAGACGCCAGTGGA |
| *NAD(P)H quinone oxidoreductase-1* (*NQO1*) | GGATTGGACCGAGCTGGAA | GAAACACCCAGCCGTCAGCTA |
| *Heme oxygenase- 1* (*HO-1*) | TTGCCAGTGCCACCAAGTTC | TCAGCAGCTCCTGCAACTCC |
| *Catalase* (*CAT*) | TTTGAGGTCACACATGACATTACCA | TCCAACGAGATCCCAGTTACCA |
| *Glutamate-cysteine ligase catalytic subunit* (*GCLC*) | GCATTATTGACGAACTGGCTACA | CTTAATCAATTTCTGGCTCACTGG |
| *Peroxisome proliferator activated receptor γ* (*PPARγ*) | CACATTACGAAGACATTCCATTCAC | GGAGATGCAGGCTCCACTTTG |
| *Nuclear factor E2-related factor-2* (*Nrf2*) | AGCCTCCAGGCAGGATTCAG | TTCCCAGTGATAAGCCCAGTTG |

with compounds, the cells were incubated for 2 h in HuMedia-KB2 containing 33 μg/mL of neutral red. The neutral red incorporated into cells was subsequently extracted using a 30% ethanol solution containing 0.1 M HCl, and cell viability was determined based on absorbance measurements (550–650 nm).

## mRNA quantification

NHEKs were cultured in 96-well plates at a density of $2.0 \times 10^4$ cells/well using HuMedia-KG2 medium and then incubated with HuMedia-KB2 containing GDS-23 or HuMedia-KB2 alone, each for 12, 18, 24, 30, 36, and 48 h. RNA was extracted using the RNeasy Mini kit (Qiagen, Hilden Germany), and cDNA was synthesized via reverse transcription using PCR Thermal Cycler Dice (Takara Bio). mRNA expression was quantified using the ΔΔCt method, which was performed using the SYBR Green Master Mix on the StepOne Real-Time PCR System (Applied Biosystems, Waltham, MA, USA). *Glyceraldehyde 3-phosphate dehydrogenase* (*GAPDH*) was used as the internal standard.

## GSH quantification

NHEKs were cultured at a density of $2.0 \times 10^4$ cells/well using HuMedia-KG2 in 96-well plates. The cells were then incubated with HuMedia-KB2 containing GDS-23 or HuMedia-KB2 alone, each for 3, 6, 24, and 48 h, and then lysed in 100 mM phosphate buffer (pH 7.4) containing 0.5% Triton X-100. Subsequently, 125 μL of 0.1 M phosphate buffer, 25 μL of 2 mM NADPH, 25 μL of 1000 U/mL glutathione (GSH) reductase, and 25 μL of cell lysate were added to the lysate, mixed, and incubated at 37˚C for 10 min. Finally, 25 μL of 10 mM Ellman's reagent was added to the lysate, and the absorbance at 412 nm was measured using Spectra-Max i3 (Molecular Devices, CA, USA). The total protein was quantified using the BCA Protein Assay Kit (Takara Bio), and the level of GSH per mg protein was calculated.

## Western blotting

NHEKs were cultured in HuMedia-KG2 medium at a density of $1.0 \times 10^5$ cells/well in 24-well plates. The cells were then incubated in HuMedia-KG2 containing GDS-23 for 12, 18, 24, 30, and 36 h, and proteins were extracted from cells using sodium dodecyl sulfate-polyacrylamide gel electrophoresis sample buffer containing 10% 2-mercaptoethanol. Protein extracts were electrophoresed on polyacrylamide gels, and proteins were transferred onto PVDF membranes using a semi-dry method. The protein levels were normalized to GAPDH (an internal standard) level.

The dilution rate and species of origin of each primary antibody are as follows. Anti-GAPDH antibody [6C5], ab8245 (Abcam, Cambridge, UK), a mouse-derived monoclonal antibody, was used at a 10,000-fold dilution. Anti-NAD(P)H-quinone oxidoreductase (NQO1) antibody [EPR3309], ab80588 (Abcam), a monoclonal antibody derived from rabbits, was used at a 5000-fold dilution. Heme oxygenase (HO-1) polyclonal antibody (Enzo Life Sciences, Farmingdale, NY, USA), a rabbit-derived polyclonal antibody, was used at a 2000-fold dilution. Catalase [H-9], sc-271803 (Santa Cruz Biotechnology, Dallas, TX, USA), a mouse-derived monoclonal antibody, was used at a 1000-fold dilution. Anti-SQSTM1/p62 antibody [2C11], ab56416 (Abcam), a mouse-derived monoclonal antibody, was used at a 10,000-fold dilution. Anti-SQSTM1/p62 (phopho S349) antibody [EPR20451], ab211324 (Abcam), a monoclonal antibody from rabbit, was used at a 2000-fold dilution. Furthermore, the secondary antibodies, rabbit anti-mouse horseradish peroxidase (HRP)-conjugated IgG H&L (HRP) (ab6728; Abcam) and HRP-conjugated goat anti-rabbit IgG H&L (ab205718; Abcam) were used at a 10,000-fold dilution.

## Nrf2 immunostaining

NHEKs were cultured at a density of $2 \times 10^4$ cells/well on a 25-mm (φ) polylysine-coated cover glass (Matsunami Glass Industry Co., Osaka, Japan) and incubated in HuMedia-KG2 for five days. Subsequently, the cells were cultured with 50 μM GDS-23-containing HuMedia-KB2 medium for 12 and 24 h and fixed with 4% formaldehyde dissolved in phosphate-buffered saline (PBS) (-). Following permeabilization with 0.05% Triton X-100 dissolved in PBS (-), the cells were treated with 10% normal goat serum (dissolved in 3% bovine serum albumin-containing PBS (-)) for 1 h at room temperature ($25 \pm 3$°C) for blocking. Subsequently, the cells were treated with 100-fold-diluted rabbit anti-Nrf2 monoclonal antibody (Abcam) overnight at 4°C for primary antibody reaction. The cells were then treated with Alexa Fluor 488 goat anti-rabbit IgG (H+L) diluted 400-fold in the dark at room temperature ($25 \pm 3$°C) for 1 h for the secondary antibody reaction. Finally, the cells were treated with 2.0 μg/mL DAPI for 10 min at room temperature in the dark for nuclear staining. After staining, the cover glass was observed under a confocal laser microscope (Olympus, Tokyo, Japan).

## p62 phosphorylation

K67 was diluted to 100 mM with dimethyl sulfoxide and then diluted with HuMedia-KB2 to prepare 25 μM K67-containing HuMedia-KB2. NHEKs were cultured in HuMedia-KG2 at a density of $2.0 \times 10^4$ cells/well in 96-well plates. Subsequently, the cells were incubated with 25 μM K67-containing HuMedia-KB2 or HuMedia-KB2 alone for 14 h, followed by incubation with GDS-23-containing HuMedia-KB2 or HuMedia-KB2 alone for 24 h. mRNA expression was quantified as described in the "mRNA quantification" section. Furthermore, NHEKs were cultured in HuMedia-KG2 medium at a density of $1.0 \times 10^5$ cells/well in 24-well plates. Subsequently, NHEKs were cultured in HuMedia-KB2 containing 25 μM K67 or HuMedia-KB2 alone for 14 h, followed by culturing in HuMedia-KB2 containing GDS-23 or HuMedia-KB2 alone for 24 h. Subsequently, protein levels were quantified as described in the aforementioned "Western Blotting" section.

## Statistical analysis

Statistical analyses were performed using the JMP Pro 16 software (SAS Institute Japan, Tokyo, Japan). Data are expressed as mean ± S.D. Dunnett's multiple comparison test was used for comparing three or more groups with the control group, and one-way analysis of variance with post-hoc Tukey test was used for multiple comparisons among three or more groups. For comparisons between two groups, the Student's sample $t$-test was used. *$p < 0.05$, **$p < 0.01$, and ***$p < 0.001$ represent statistical significance.

# Results

## Effect of GDS-23 on the expression of antioxidant proteins

GDS-23, a diacylglycerol PEG adduct, induced a significant decrease in cell viability at a concentration of 200 μM in NHEKs. Although not significant, an approximatey 10% decrease in cell viability was observed at a concentration of 100 μM GDS-23. In contrast, no cytotoxicity was observed at concentrations of 50 μM or less (S1 Fig). Based on these findings, the maximum concentration of GDS-23 was set at 50 μM in this study. The protective effect of GDS-23 against oxidative stress was evaluated by measuring the mRNA expression of antioxidant proteins in NHEKs. The mRNA expression of several antioxidant protein-encoding genes such as *HO-1* (*HMOX1*), *NQO1*, *glutamate-cysteine ligase catalytic subunit* (*GCLC*), and *catalase* significantly increased after GDS-23 treatment compared to the control (Fig 2). Moreover, the

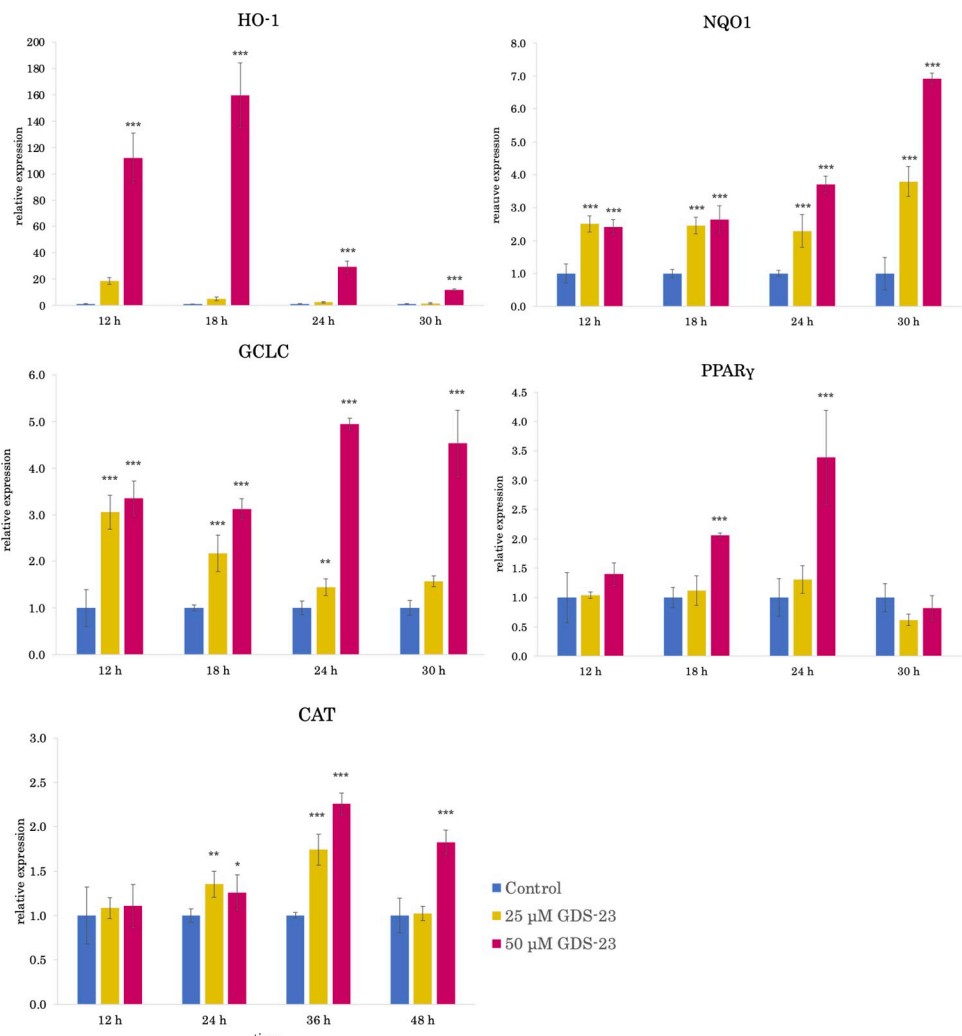

**Fig 2. Effects of GDS-23 on the expression of genes involved in antioxidative stress responses.** NHEKs were treated with either 25 or 50 μM GDS-23 for 12, 18, 24, 30, 36, and 48 h. Control cells were treated with vehicle. mRNA levels of *HO-1*, *NQO1*, *GCLC*, *CAT*, and *PPARγ* were analyzed using quantitative polymerase chain reaction. Results are expressed as mean ± standard deviation (n = 4). Significance; *p < 0.05, **p < 0.01, ***p < 0.001 vs. control cells (Dunnett's test). GDS-23, glyceryl polyethylene glycol-23 distearate; NHEK, normal human epidermal keratinocyte; HO-1, heme oxygenase-1; NQO1, NAD(P)H-quinone oxidoreductase 1; GCLC, glutamate-cysteine ligase catalytic subunit; CAT, catalase; PPARγ, peroxisome proliferator-activated receptor γ.

mRNA expression of *PPARγ* (Fig 2) and *Nrf2* (S2 Fig) significantly increased compared to the control. Since the mRNA expression of several intracellular antioxidant proteins was induced, the level of each protein was also evaluated. HO-1 protein level significantly increased after 12 h of treatment with GDS-23, and NQO1 and catalase protein levels increased after 18 h of treatment compared to the control (Figs 3A–3C and S3).

## Effect of GDS-23 on intracellular GSH level

As GDS-23 significantly upregulated the expression of genes encoding antioxidant proteins such as GSH synthetase, we measured the level of GSH, which plays a major role in intracellular antioxidant activity. The GSH level in the cells significantly increased in a dose-dependent manner, and it was gradually increased after 24 h of GDS-23 treatment (Fig 4). In particular,

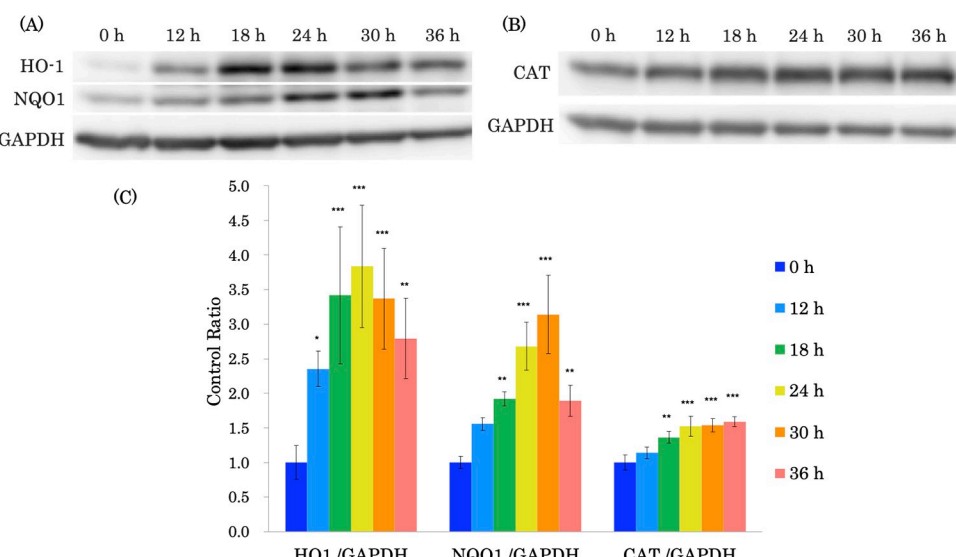

**Fig 3. Effect of GDS-23 on the levels of proteins involved in antioxidative stress responses in NHEKs.** (A and B) Cells were treated with 50 μM GDS-23 for 12, 18, 24, 30, and 36 h, and the levels of HO-1, NQO1, and CAT proteins were determined using western blotting. (C) Expression levels of each protein were normalized to the level of GAPDH and expressed as a ratio to control cells. Results are expressed as mean ± standard deviation (n = 4). Significance; *p < 0.05, **p < 0.01, ***p < 0.001 vs. control cells (Dunnett's test). GDS-23, glyceryl polyethylene glycol-23 distearate; NHEK, normal human epidermal keratinocyte; HO-1, heme oxygenase 1; NQO1, NAD(P)H-quinone oxidoreductase 1; CAT, catalase; GAPDH, glyceraldehyde 3-phosphate dehydrogenase.

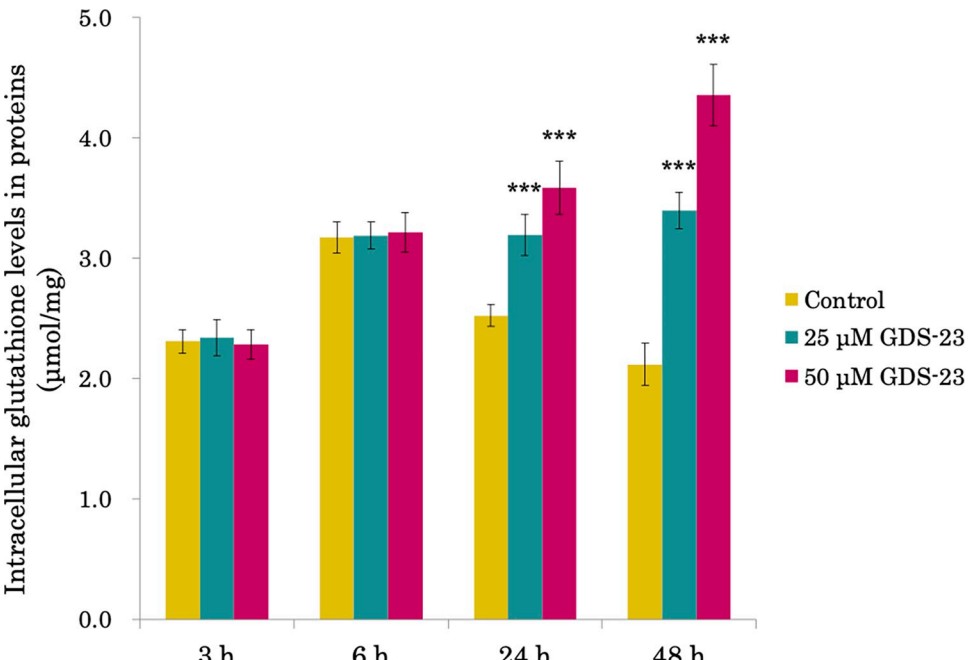

**Fig 4. Effect of GDS-23 on intracellular glutathione level in NHEKs.** The cells were treated with 25 or 50 μM GDS-23 for 3, 6, 24, and 48 h. Intracellular glutathione level was quantified and is presented after normalization with protein level. Results are expressed as mean ± standard deviation (n = 6). Significance; ***p < 0.001 vs. control cells (Dunnett's test). GDS-23, glyceryl polyethylene glycol-23 distearate; NHEK, normal human epidermal keratinocyte.

48 h after treatment with 50 μM GDS-23, the GSH level was approximately twice that of the control.

## Nrf2 activation by GDS-23

Upon activation, Nrf2 is released via Keap1-dependent proteosomal degradation in the cytoplasm, stabilized, and translocated to the nucleus. We examined the activation state of Nrf2 by evaluating the localization of Nrf2 via fluorescence immunostaining. For cells treated with GDS-23 for 12 and 24 h, strong fluorescence was observed in the nucleus, clearly indicating that Nrf2 was stabilized and translocated to the nucleus following this treatment (Fig 5A).

## Role of phosphorylated p62 in GDS-23-mediated Nrf2 activation

The gene expression of antioxidant proteins is regulated by Nrf2. Nrf2 activation is mostly triggered by the oxidation of the SH group of Keap1; however, the chemical structure of GDS-23

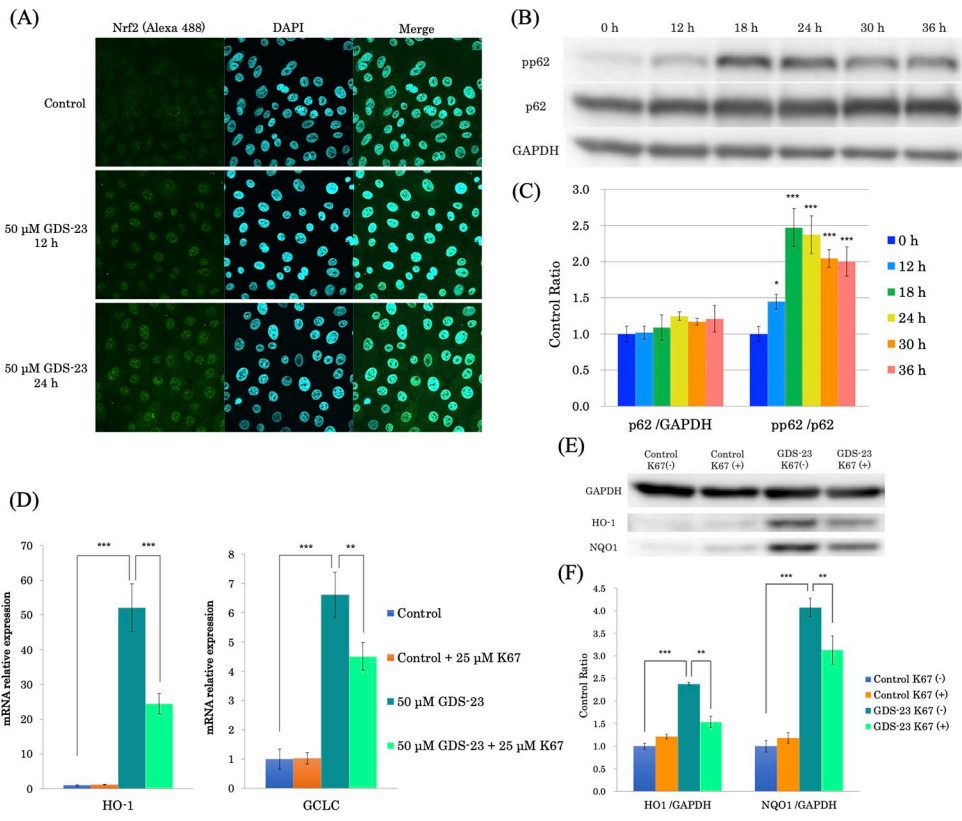

**Fig 5. Role of phosphorylated p62 in GDS-23-mediated Nrf2 activation in NHEKs.** (A) Subcellular localization of Nrf2 in NHEKs treated with 50 μM GDS-23 for 12 and 24 h. The cells were treated with 50 μM GDS-23 for 12 or 24 h and then immunostained for Nrf2. (B, C) Effect of GDS-23 on p62 and pp62 protein levels in NHEKs. (D) NHEKs were treated with 50 μM GDS-23 for 12, 18, 24, 30, and 36 h. p62 and pp62 protein levels were determined using western blotting. (C) p62 and pp62 bands in western blots were normalized using GAPDH and p62, respectively, and expressed as a ratio to the control cells. Data are presented as mean ± standard deviation. (n = 4). Significance; $*p < 0.05$, $**p < 0.01$, $***p < 0.001$ vs. control (Dunnett's test). (D–F) Effect of K67 on HO-1 and GCLC expression by GDS-23. (D) NHEKs were pretreated with K67 and then treated with GDS-23 for 24 h. *HO-1* and *GCLC* expression levels were quantified using quantitative polymerase chain reaction. (E and F) NHEKs were pretreated with K67 and then treated with GDS-23 for 24 h. HO-1 and NQO1 proteins were detected using (E) western blotting and normalized using (F) GAPDH. Results are expressed as mean ± standard deviation (n = 4). Significance; $**p < 0.01$, $***p < 0.001$ (Tukey's test). GDS-23, glyceryl polyethylene glycol-23 distearate; NHEK, normal human epidermal keratinocyte; GAPDH, glyceraldehyde 3-phosphate dehydrogenase; Nrf2, nuclear factor E2-related factor-2; pp62, phosphorylated p62; HO-1, heme oxygenase 1; NQO1, NAD(P)H-quinone oxidoreductase 1; GCLC, glutamate-cysteine ligase catalytic subunit.

does not reflect electrophilic properties, suggesting that a different mechanism may be involved. Recently, it was reported that the phosphorylation of p62 at Ser349, an autophagy-related adaptor protein [18], increases its binding to Keap1, causing Nrf2 to detach from Keap1 and become stable [19]. Therefore, we similarly examined the role of p62 in the Nrf2-activating effect of GDS-23 in the present study.

GDS-23 treatment did not alter the protein expression of p62 compared to the control; however, the phosphorylation of p62 at Ser349 significantly increased after 12 h of treatment compared to the control (Figs 5B, 5C and S4). Thus, GDS-23 induced the phosphorylation of p62 early after treatment. The binding affinity of phosphorylated p62 and Keap1 is considerably high, and Nrf2 does not bind to Keap1 and translocate to the nucleus, which induces the transcriptional activation of the target gene [19]. Therefore, we investigated the effect of K67, which inhibits the protein–protein interaction between Keap1 and phosphorylated p62 [22], on GDS-23-induced Nrf2-dependent gene expression of antioxidant proteins. The addition of GDS-23 to NHEKs cultured in the presence of K67 for 14 h decreased the induction of *HMOX1* and *GCLC* expression (Fig 5D). Moreover, there was a decrease in the levels of HO-1 and NQO1 proteins, which were enhanced by treatment with GDS-23 (Figs 5E, 5F and S4).

### Protective effect of GDS-23 against hydroquinone-induced cytotoxicity

Hydroquinone is one of the most effective agents used to treat melanopathy [23]; however, it is quickly oxidized and converted to highly toxic *p*-benzoquinone and hydroxybenzoquinone, which results in topical toxicity [24]. Based on the aforementioned results, we evaluated the possibility of attenuating the toxicity of hydroquinone by GDS-23 treatment, which was found to activate Nrf2 and enhance the intracellular antioxidant system. We also evaluated the toxicity of hydroquinone when its stability was enhanced with sodium pyrosulfite, which has a reduction action. The cells pretreated with 25 or 50 μM GDS-23 for 24 h showed a significantly improved viability after 200 μM hydroquinone treatment compared to those without pretreatment (S5 Fig). The effect of GDS-23 was found to be comparable to that in experimental models wherein hydroquinone oxidation was prevented by sodium pyrosulfite (Fig 6).

### Discussion

For the present study, we used GDS-23, which had the highest antioxidant activity against keratinocytes among similar compounds based on preliminary studies. Although we should have used an inactive structural analog as the control compound, all of our currently available two-chain diacylglycerol PEG adducts showed the activity, so we decided to use GDS-23 in this study to investigate the mechanism of Nrf2 activation and its efficacy on keratinocytes. Structure-activity relationships, including inactive forms, will be the subject of future studies. Upon activation of the Nrf2 signaling pathway, multiple antioxidant enzymes and proteins that attenuate oxidative stress, such as HO-1, NQO1, and GCLC, are induced in the cells [25]. Our findings suggest that Nrf2 is activated by GDS-23 owing to the upregulated expression of multiple antioxidant proteins such as HO-1, NQO1, and GCLC and finally the increased intracellular GSH contents. Peak intracellular GSH levels were observed 48 h after GDS-23 treatment, which appears to lag behind the GCLC gene expression, which peaked 24 h after treatment. However, this is likely because peak protein expression follows gene expression and peak metabolites emerge later. Furthermore, GSH levels are regulated by multiple steps, including the cystine uptake transporter xCT in addition to GCLC [26]. GDS-23 treatment stabilized Nrf2 in the nucleus. In general, Nrf2 is activated when cells are exposed to oxidative stress or electrophiles, which oxidizes the thiol group of Keap1, causing Nrf2 to detach from Keap1 and translocate to the nucleus to activate transcription of the target genes [27]. However, GDS-23

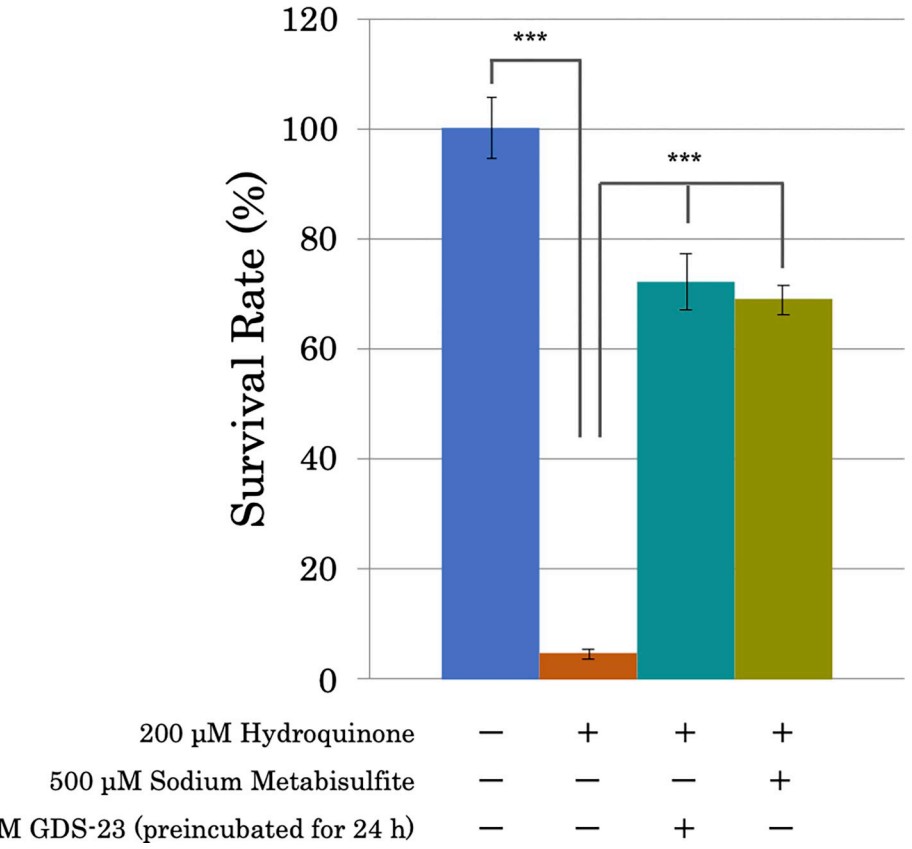

**Fig 6. Effect of GDS-23 on hydroquinone-induced cytotoxicity.** NHEKs were pretreated with 25 μM GDS-23 for 24 h and then treated with 200 μM hydroquinone. Furthermore, the cytotoxic effects of 200 μM hydroquinone alone and 200 μM hydroquinone in combination with 500 μM sodium pyrosulfite were compared. Cytotoxicity was measured using the neutral red assay and expressed as a percentage relative to the survival of the control cells. Results are expressed as mean ± standard deviation (n = 6). Significance; ***p < 0.001 (Tukey's test). GDS-23, glyceryl polyethylene glycol-23 distearate; NHEK, normal human epidermal keratinocyte.

is not electrophilic; thus, a different pathway for Nrf2 activation mechanism was considered. In this study, we focused on the pathway wherein the phosphorylation of p62 increases its binding affinity to Keap1, causing Nrf2 to detach from Keap1 and become stable [28]. p62 contains multiple domains that mediate interactions with various bonds and functions as a hub for diverse signaling events such as amino acid sensing and oxidative stress responses [29], in addition to functioning as a selective autophagy-related protein that degrades ubiquitinated substrates [30]. Furthermore, p62 is upregulated by various stressors. Under oxidative stress conditions, a positive feedback mechanism is triggered wherein p62 is induced by Nrf2 activation and contributes to Nrf2 activation [31].

In this study, we revealed that GDS-23 promotes the phosphorylation of p62 at Ser349, and this was accompanied by an increase in mRNA and protein expression of antioxidant proteins. The peak p62 phosphorylation was observed 18 h after treatment, followed by the peak gene expression for *HO-1* and *NQO1* which were observed 18 and 30 h after treatment, respectively. These results are consistent with a time course in which p62 activation may have caused Nrf2-dependent gene expressions. Our findings suggest that the p62 phosphorylation induced by GDS-23 is involved in the Nrf2 activation mechanism. Additionally, an increase in Nrf2 mRNA has been observed 24 h after GDS-23 treatment (S2 Fig), which is believed to be

attributed to the positive feedback mechanism of the p62-Keap1-Nrf2 pathway. However, the induction level of Nrf2 was only 1.4-fold, and, therefore, the activation of Nrf2 by GDS-23 would be primarily due to the stabilization of Nrf2. This stabilization could be related to the role of p62, which depending on the phosphorylation site, can exhibit increased affinity for polyubiquitin chains leading to autophagy-assisted degradation [32,33] or increased affinity for Keap1 [28]. In the latter case, the newly synthesized Nrf2 is not bound by Keap1, and it translocates to the nucleus, thereby triggering transcriptional activation [34]. The fact that the p62 protein level did not decrease despite the phosphorylation of p62 in this study suggests that a positive feedback mechanism may exist, in which Nrf2 activation leads to p62 production. In addition, phosphorylated p62 binds to Keap1 to form protein aggregates, which are degraded by autophagy via interaction with LC3, an autophagosomal marker protein [19,35]. Based on these findings, the decrease in the production of phosphorylated p62 after a peak at 18–24 h after treatment with GDS-23 was attributed to its degradation by selective autophagy. Nrf2 immunostaining showed strong fluorescence in cell nuclei after 12 and 24 h of GDS-23 treatment, suggesting that the nuclear translocation of Nrf2 is triggered after p62 phosphorylation. Furthermore, treatment with K67, an inhibitor of the Keap1-phosphorylated p62 protein–protein interaction [22], significantly inhibited the increase in the mRNA and protein levels of antioxidant proteins. These results support the function of phosphorylated p62 as one of the mechanisms of Nrf2 activation induced by GDS-23. However, the effect of K67 was partial, and the activation pathway of Nrf2 is complex; thus, the activation of Nrf2 via GDS-23 treatment may not be solely due to the phosphorylation of p62.

PPARs constitute a family of three nuclear receptors and transcription factors that regulate gene expression and function as lipid sensors; they also play a role in the regulation of metabolism and inflammation [36]. PPARγ has been recently reported as a transcription factor, which along with Nrf2, plays important role in the cellular antioxidant defense system [37]. Furthermore, several studies suggest that the Nrf2 and PPARγ pathways are mutually regulated by enhancing each other's protein expression [20,37]. The knockdown of *Nrf2* decreases PPARγ expression [38], whereas *PPARG* knockdown decreases Nrf2 expression [21,39]. This study showed that *PPARG* was significantly upregulated in GDS-23-treated cells. Therefore, it is possible that PPARγ expression is partially involved in the activation of Nrf2 by GDS-23. Future studies should comprehensively investigate this subject.

The present study indicates that GDS-23 can enhance the intracellular antioxidation system without inducing oxidative stress in cells, and it may inhibit and prevent UV-induced pigmentation, skin cancer, and skin aging caused by oxidative stress. Furthermore, as GDS-23 itself enhances the intracellular antioxidant system, proper niosome formation leading to skin penetration may induce antioxidant effects. Therefore, GDS-23 is expected to function not only as a DDS carrier in the form of a niosome but also as a multifunctional material for DDS, which enhances antioxidant function by migrating to the skin. Nevertheless, the impact of the drug encapsulated in the niosome during the action of GDS-23 depends on the specific drug, which warrants further investigation. We are currently conducting a study that more closely mimics actual skin tissue conditions by utilizing a 3-dimensional tissue culture for efficacy verification.

Additionally, we also evaluated the effects of GDS-23 on hydroquinone-induced cytotoxicity [40] as an application of multifunctional DDS material. Hydroquinone is an effective skin-whitening ingredient used to treat hyperpigmentation [41]; however, it is easily oxidized to highly toxic *p*-benzoquinone and hydroxybenzoquinone. Therefore, cellular injury caused by hydroquinone may be attributed to the oxidative byproducts [42]. Intracellularly, *p*-benzoquinone and hydroxybenzoquinone formed by oxidation can be reduced by NQO1, a two-electron reductase, to hydroquinone, which is less toxic than the oxidation products [43]. Thus,

we hypothesized that increasing the intracellular level of antioxidant proteins such as NQO1 may reduce the cytotoxicity of chemicals such as hydroquinone, which show increased toxicity when oxidized. During our investigation, we found that the cytotoxicity of hydroquinone was attenuated in cells pretreated with GDS-23, which enhances the intracellular antioxidant system. It is considered that NQO1 and several other antioxidant proteins produced in the cells owing to the enhanced intracellular antioxidant system prevented the oxidation of hydroquinone or induced the reduction of benzoquinone. This finding suggests that GDS-23 can attenuate the intracellular toxicity of hydroquinone while eliciting its anti-pigmentation and skin-whitening effects, and it has potential application as an additive for therapeutic drugs and cosmetics.

## Conclusions

In this study, we found that GDS-23 activates Nrf2 and the intracellular antioxidant system. Furthermore, one of the mechanisms of Nrf2 activation involves the phosphorylation of p62, an autophagy-related adaptor protein. In addition, GDS-23 attenuated intracellular oxidative stress and toxicity of chemicals, which are toxic when oxidized in the body, without causing oxidative stress in cells. Based on these results, we conclude that GDS-23, a PEG-lipid, can form niosomes and facilitate transdermal penetration of drugs. It can also be applied as a "multifunctional DDS material" that enhances internal antioxidant systems and reduces oxidative stress spontaneously.

## Supporting information

**S1 Fig. Evaluation of GDS-23 cytotoxicity and protein quantification in keratinocytes.** NHEKs were treated with GDS-23 at concentrations ranging from 12.5 to 200 μM for 24 h, and cytotoxicity was evaluated using a neutral red assay. Results are expressed as a percentage survival compared to the control cells and as mean ± S.D. (n = 6). Significance; $^{**}p < 0.01$, $^{***}p < 0.001$ vs. control (Dunnett's test). GDS-23, glyceryl polyethylene glycol-23 distearate; NHEK, normal human epidermal keratinocyte.
(TIF)

**S2 Fig. Effects of GDS-23 on Nrf2 gene expressions.** NHEKs were treated with 50 μM GDS-23 for 12 and 24 h. Control cells were treated with vehicle. *Nrf2* mRNA levels were analyzed using qPCR. Results are expressed as mean ± S.D. (n = 4). Significance; $^{*}p < 0.05$ vs. control cells (Student's *t*-test). GDS-23, glyceryl polyethylene glycol-23 distearate; NHEK, normal human epidermal keratinocyte; Nrf2, nuclear factor E2-related factor-2.
(TIF)

**S3 Fig. Raw images of Fig 3.** (A) To simultaneously or sequentially detect antigens using multiple antibodies, the membrane was cut below the 50 kDa marker. This was done to specifically focus on the detection of GAPDH (36 kDa), HO-1 (32 kDa), and NQO1 (31 kDa). The detection was performed first for HO-1, followed by NQO1, and finally GAPDH. After each detection, the antibodies were stripped from the membrane and the process was repeated for the next protein of interest. (B) To simultaneously detect antigens using multiple antibodies, the membrane was cut at two positions: below the 50 kDa marker and below the 75 kDa marker. This was done to specifically focus on the detection of GAPDH (36 kDa) and CAT (64 kDa). Red squares indicate the band position used in Fig 3A and 3B.
(TIF)

**S4 Fig. Raw images of Fig 5.** (B) To simultaneously or sequentially detect antigens using multiple antibodies, the membrane was cut at two line positions: below the 50 kDa marker and

below the 75 kDa marker. This was done to specifically focus on the detection of GAPDH (36 kDa), p62 (62 kDa), and phosphorylated p62 (62 kDa). The detection was performed first for p62 and then for phosphorylated p62. After each detection, the antibodies were stripped from the membrane and the process was repeated for the next protein of interest. (E) To simultaneously or sequentially detect antigens using multiple antibodies, the membrane was cut below the 50 kDa marker. This was done to specifically focus on the detection of GAPDH (36 kDa), HO-1 (32 kDa), and NQO1 (31 kDa). The detection was performed first for HO-1, followed by NQO1, and finally GAPDH. After each detection, the antibodies were stripped from the membrane and the process was repeated for the next protein of interest. Red squares indicate the band position used in Fig 5B and 5E.
(TIF)

**S5 Fig. Effect of GDS-23 on hydroquinone-induced cytotoxicity.** NHEKs were pretreated either with vehicle or GDS-23 (25 or 50 μM) for 24 h and subsequently treated with 200 μM hydroquinone. Cytotoxicity was measured using the neutral red assay and expressed as a percentage relative to the control cells. Results are expressed as mean ± S.D. (n = 6). Significance; ***$p < 0.001$ (Tukey's test). GDS-23, glyceryl polyethylene glycol-23 distearate; NHEK, normal human epidermal keratinocyte.
(TIF)

## Acknowledgments

The experiments were conducted at the Showa University Pharmacological Research Center. We would like to thank Ms. Miyu Takahashi, Ms. Haruka Mukai, Ms. Maria Tanaka, Ms. Sui Nakagawa, Ms. Yui Yamamoto, and Ms. Hitomi Mogi at Showa University for their support in facilitating the research.

## Author Contributions

**Conceptualization:** Tatsuro Miyoshi.

**Data curation:** Tatsuro Miyoshi.

**Formal analysis:** Tatsuro Miyoshi, Brian C. Keller, Takashi Ashino.

**Investigation:** Tatsuro Miyoshi.

**Methodology:** Brian C. Keller, Takashi Ashino, Satoshi Numazawa.

**Project administration:** Tatsuro Miyoshi.

**Supervision:** Satoshi Numazawa.

**Writing – original draft:** Tatsuro Miyoshi.

**Writing – review & editing:** Satoshi Numazawa.

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
