## [Decision Letter · Decision Letter 0]

10 Jul 2023

PONE-D-23-15747Noncanonical mechanism of Nrf2 activation by diacylglycerol polyethylene glycol adducts in normal human epidermal keratinocytesPLOS ONE

Dear Dr. Miyoshi,

Thank you for submitting your manuscript to PLOS ONE. After careful consideration, we feel that it has merit but does not fully meet PLOS ONE’s publication criteria as it currently stands. Therefore, we invite you to submit a revised version of the manuscript that addresses the points raised during the review process.

We look forward to receiving your revised manuscript.

Kind regards,

Arunava Roy, Ph.D.

Academic Editor

PLOS ONE

Journal Requirements:

"This study was funded by Beverly Glen Laboratories, Inc. https://www.bglen.com

TM is an employee of Beverly Glen Laboratories, Inc. BCK is Chief Science Officer of Beverly Glen Laboratories, Inc.

TM conceived the idea for this study, conducted the experiments, and wrote the paper. BCK contributed to analysis of the results."

"I have read the journal's policy and the authors of this manuscript have the following competing interests.

T. Miyoshi is an employee of Beverly Glen Laboratories, Inc. Brian C. Keller is Chief Science Officer of Beverly Glen Laboratories, Inc. S. Numazawa and T. Ashino serves as a member of a joint research project between Beverly Glen Laboratories, Inc. and Showa University. The authors declare that they have no known competing financial interests or personal relationships that could have appeared to influence the work reported in this paper."

6. We note that Figure 7 in your submission contain copyrighted images. All PLOS content is published under the Creative Commons Attribution License (CC BY 4.0), which means that the manuscript, images, and Supporting Information files will be freely available online, and any third party is permitted to access, download, copy, distribute, and use these materials in any way, even commercially, with proper attribution. For more information, see our copyright guidelines: http://journals.plos.org/plosone/s/licenses-and-copyright.

a. You may seek permission from the original copyright holder of Figure 7to publish the content specifically under the CC BY 4.0 license. 

Reviewers' comments:

Reviewer's Responses to Questions

**Comments to the Author**

1. Is the manuscript technically sound, and do the data support the conclusions?

Reviewer #1: Partly

Reviewer #2: Yes

2. Has the statistical analysis been performed appropriately and rigorously? 

Reviewer #1: Yes

Reviewer #2: Yes

3. Have the authors made all data underlying the findings in their manuscript fully available?

Reviewer #1: Yes

Reviewer #2: Yes

4. Is the manuscript presented in an intelligible fashion and written in standard English?

Reviewer #1: Yes

Reviewer #2: Yes

5. Review Comments to the Author

Reviewer #1: The study shows ease of delivery and compatibility with several water and oil-soluble drugs with GDS-23 PEG niosomes and their role in p62-Nrf-2 dependent antioxidant effects, with potential protective effects on Skin.

In my opinion, the study concept is novel, However, there are some concerns that need to be addressed. The followings are my comments:

1. The authors have chosen 25uM and 50uM doses of GDS-23. Have they gone higher? What is the permissible maximum dose, given the fact that over-activation of antioxidant activity could be toxic.

2. Are there specific drugs compatible for delivery with these niosomes? How would the packing capacity of niosomes positively or negatively impact the level of antioxidant effects?

3. Are there any studies on tissue penetrance of niosomes ? Please incorporate in discussions.

4. The antioxidants analyzed for mRNA and protein expression in response to GDS-23 expression in keratinocytes have different time course for peak activity. P62 phosphorylation peaks at 18 hours post GDS-23 treatment. GSH activity peaks at 48 hours post treatment . With these differences in the time points of activation peak, how do the authors correlate the cause-effect function of GDS-23-pP62-Nrf-2 axis with the executioners of antioxidant signaling ?

5. Nrf-2 antibody staining, shows nuclear translocation in GDS-23 groups. Does 24 hour treatment show enhanced nuclear translocation compared to 12 hours? Does GDS-23 increase nrf2 mRNA expression along with nuclear translocation? Is there any dose dependent difference in the level of nuclear translocation of Nrf2? Quantification of these data for nuclear and cytosolic Nrf-2 levels would be necessary to conclude. Additionally, the signal intensity for Nrf-2 in the representative images are not very strong. A no-primary, secondary antibody control for all the groups would be beneficial.

6. Excess antioxidant activity has been shown to cause toxicity. Hydrquinone is used for its anti-pigmentation effects and it exerts its effects by reducing melanin. Melanocytes have high antioxidant level. High melanin containing skin already has higher Nrf2 activity. Could the authors comment on potential over-activation effects of antioxidant properties, causing any side effects of cytotoxicity in those cells? In vitro studies in cells such as Melanocytes with Hydroquinone would be helpful.

7. With the experiment involving Hydroquinone, authors have incorporated 25uM dose of GDS-23. Since, the western blot data are all shown in 50uM dose of GDS-23 it would be important to show the effect of both doses on the cytotoxity and survival of cells. Similar experiment in melanin-containing cells would be necessary to eliminate negative impact of excessive anti-oxidant properties if any.

Reviewer #2: The article titled “Noncanonical mechanism of Nrf2 activation by diacylglycerol polyethylene

glycol adducts in normal human epidermal keratinocytes” by Miyoshi et. al. discusses the role

of GDS-23, a PEG adduct, as a mediator of activation of antioxidation pathway via NRF-2

pathway. The study is purely based on in vitro experiments and findings are significant. The

scientific concept is fresh, and experiments are executed well.

My comments on the manuscript are below:

1. The article shows the anti-oxidative effect of GDS-23 in a convincing manner.

However, the authors haven’t used any structural controls in the studies. Using a

different PEG adduct that is structurally similar to GDS-23 is necessary to ensure the

observed effect is specific to GDS-23.

2. Since all the experiments are performed in vitro, there is no information on the toxicity

of GDS-23. A murine model with/without treatment with GDS-23 can address that. It is

also important to know if there are off target effects of the PEG lipid.

3. Apart from Nrf-2, what other factors are involved on the GDS-23 treatment? RNAseq

of cells treated with/without GDS-23 can be performed to understand the global effect

of the lipid.

4. Antioxidant effect of Nrf-2 is known to influence mitochondrial functions. Did the

authors check if there is the effect of GDS-23 on mitochondria?

Minor comment:

1. Authors are suggested to keep the nomenclature same across the text and figures.

E.g., Sodium pyrosulfite and Sodium metabisulphite are interchangeably used in text

and fig 6.

6. PLOS authors have the option to publish the peer review history of their article (what does this mean?). If published, this will include your full peer review and any attached files.

Reviewer #1: No

Reviewer #2: No

---

## [Author Response · Author response to Decision Letter 0]

9 Aug 2023

Reviewer #1:

Q1-1. The authors have chosen 25uM and 50uM doses of GDS-23. Have they gone higher? What is the permissible maximum dose, given the fact that over-activation of antioxidant activity could be toxic.

A1-1.

The concentration of GDS-23 solution typically used on the skin is around 1-2% (approximately 6-12 mM). In skin permeation tests using a three-dimensional cultured skin model, it has been demonstrated that 0.5-2% of applied GDS-23 penetrates the skin when a 2% GDS-23 solution is used. Considering this permeability, about 50 μM GDS-23 would theoretically surround dermal keratinocytes, which falls within the concentration range currently being tested in this study. Additionally, when cells were treated with 100 μM GDS-23 for 24 hours, although there was no significant difference, the survival rate of the cells dropped by 5-10% (S1 Fig.). On the other hand, at concentrations of 50 μM or less, the survival rate was equivalent to the control cells. Based on these findings, the maximum concentration was set at 50 μM in this study, as it does not affect cell toxicity. If there is excessive antioxidant activity, it is thought to occur at concentrations of 100 μM or more. The revisions regarding the cellular toxicity of GDS-23 in NHEK were made on page 13, lines 212 – 218 of the manuscript.

Q1-2. Are there specific drugs compatible for delivery with these niosomes? How would the packing capacity of niosomes positively or negatively impact the level of antioxidant effects?

A1-2

While there are no specific drugs identified as particularly compatible, our preliminary results have shown that GDS-23 treatment can amplify the gene expression of transporters such as SVCT2 and xCT, which facilitate the uptake of vitamin C and cystine into cells, respectively. Considering the possible synergistic effects, ascorbic acid, cysteine, or cystine could be suitable candidates. These results are still preliminary at this stage, so we have decided not to include them in this paper.

As shown in this study, GDS-23 itself enhances the intracellular antioxidant system, it is possible that the penetration (absorption) into the skin through appropriate niosome formation would induce the antioxidant effects. We believe that the packing capacity of the niosomes does not have a positive or negative impact on the antioxidative effect. Whether the encapsulated drug affects the action of GDS-23 is dependent on the drug, and this is a subject for future investigation. The revisions in these points were made in the Discussion on page 24, lines 416 - 422 of the manuscript.

Q1-3. Are there any studies on tissue penetrance of niosomes ? Please incorporate in discussions.

A1-3.

As explained in the Introduction (page 3, lines 49 - 52), we have previously reported that GDS-23 readily forms niosomes with multiple membranes and enhances the skin penetration of drugs. In terms of tissue penetrance, studies using a three-dimensional cultured epidermal model have been conducted, and obtained preliminary findings suggest that GDS-23 increases penetrance by approximately 2.5 times. These studies are still in the preliminary stages, and we plan to present the details of these experiments in a subsequent paper.

Q1-4. The antioxidants analyzed for mRNA and protein expression in response to GDS-23 expression in keratinocytes have different time course for peak activity. P62 phosphorylation peaks at 18 hours post GDS-23 treatment. GSH activity peaks at 48 hours post treatment . With these differences in the time points of activation peak, how do the authors correlate the cause-effect function of GDS-23-pP62-Nrf-2 axis with the executioners of antioxidant signaling ?

A1-4.

The phosphorylation peak of p62 occurred 18 hours post-treatment, the gene expression peak of HO-1 was observed at 24 hours, and that of NQO-1 was at 30 hours. Therefore, there is no so much difference in terms of the time course of p62 activation and Nrf2-dependent gene expression. On the other hand, the peak of intracellular glutathione observed at 48 hours, which occurred later than the gene expressions, can be explained by the general trend that the peak of gene expression is followed by the peak of protein expression, which is then followed by the peak of metabolic products. Furthermore, proteins, such as GCLC and the cystine uptake transporter xCT, are involved in the synthesis of glutathione, thereby regulating the intracellular GSH levels in multiple steps. In the present study, the gene expression peak of GCLC was observed from 24 to 30 hours after treatment. Assuming that GSH synthesis increased from there, it is consistent with a peak in intracellular GSH levels occurring at 48 hours. The revisions regarding these issues were made in the Discussion on page 20, lines 345-350 and page 21, lines 366 – 369 of the manuscript.

Q1-5. Nrf-2 antibody staining, shows nuclear translocation in GDS-23 groups. Does 24 hour treatment show enhanced nuclear translocation compared to 12 hours? Does GDS-23 increase nrf2 mRNA expression along with nuclear translocation? Is there any dose dependent difference in the level of nuclear translocation of Nrf2? Quantification of these data for nuclear and cytosolic Nrf-2 levels would be necessary to conclude. Additionally, the signal intensity for Nrf-2 in the representative images are not very strong. A no-primary, secondary antibody control for all the groups would be beneficial.

A1-5.

While it is true that strong fluorescence was observed in the nucleus 12 and 24 hours after GDS-23 treatment, the term "time-dependent" may lead to misunderstanding. We have revised the manuscript to state that "Strong fluorescence was observed in the nucleus 12 and 24 hours after GDS-23 addition, indicating that Nrf2 was stabilized and translocated to the nucleus" (page 16, lines 266-269). We are confident in the activation of the Nrf2 antioxidant system, as evidenced by the increased gene expression and protein levels of downstream elements such as HO-1 and NQO1, along with the nuclear translocation of Nrf2 observed by immunostaining. Furthermore, the amplification of Nrf2 mRNA was observed 24 hours later (Fig. S2), which could be due to the positive feedback mechanism of the p62-Keap1-Nrf2 pathway. However, the induction level of Nrf2 was only 1.4 times, and we believe that Nrf2 activation was mainly due to its stabilization. We have added a note on these points in the Results (page 16, lines 266 – 269) and the Discussion (page 22, lines 371 – 378).

Q1-6. Excess antioxidant activity has been shown to cause toxicity. Hydrquinone is used for its anti-pigmentation effects and it exerts its effects by reducing melanin. Melanocytes have high antioxidant level. High melanin containing skin already has higher Nrf2 activity. Could the authors comment on potential over-activation effects of antioxidant properties, causing any side effects of cytotoxicity in those cells? In vitro studies in cells such as Melanocytes with Hydroquinone would be helpful.

A1-6.

We have conducted preliminary studies with melanocytes using GDS-23 and hydroquinone in a monolayer cell culture. These results indicated that GDS-23 attenuates cellular damage caused by benzoquinone, which is produced by the oxidation of hydroquinone (References 11). The concentrations of GDS-23 used in these studies were much higher than those used in the current study and did not cause damage to the melanocyte cells (X1 Fig, as illustrated below). Therefore, we believe that at concentrations where GDS-23 does not induce cellular damage, it does not exhibit side effects (excessive antioxidant activity) in melanocytes. 

The focus of this paper, however, is the pharmacological action of GDS-23 in keratinocytes, the outermost living cells of the skin, where the initial reaction occurs when drugs permeate the skin. Because melanocytes are pigment cells located deeper than keratinocytes, a more complex scenario that takes into account the three-dimensional structure of skin tissue must be presented in order to understand their inherent reactivity. Therefore, we have decided not to include these points in the current paper. 

Q1-7. With the experiment involving Hydroquinone, authors have incorporated 25uM dose of GDS-23. Since, the western blot data are all shown in 50uM dose of GDS-23 it would be important to show the effect of both doses on the cytotoxity and survival of cells. Similar experiment in melanin-containing cells would be necessary to eliminate negative impact of excessive anti-oxidant properties if any.

A1-7.

GDS-23 at 25 μM and 50 μM equivalently decreased hydroquinone-induced cellular damage (S5 Fig). GDS-23, within the concentration range of 25~50 μM, could sufficiently enhance intracellular antioxidant activity, contributing to the mitigation of hydroquinone-induced cellular damage. The revisions were made on page 20, lines 335-341 of the manuscript.

Reviewer #2: 

Q2-1. The article shows the anti-oxidative effect of GDS-23 in a convincing manner.

However, the authors haven’t used any structural controls in the studies. Using a

different PEG adduct that is structurally similar to GDS-23 is necessary to ensure the observed effect is specific to GDS-23.

A2-1. Compounds related to GDS-23, such as those with a PEG chain length of 12 or those with a PEG chain length of 12 and a change in fatty acid from hydroxystearic acid to myristic acid, have been found to exhibit similar effects (References 11). However, at present, we do not have data to demonstrate whether this effect is specific to the structural family. To investigate the structural specificity, we are currently considering a comparison with compounds that have a single-chain fatty acid and are structurally similar to GDS-23. As of now, we do not have data available to indicate whether there is structural specificity. The revisions regarding this issue were made in the Discussion on page 19, lines 320 – 322 of the revised manuscript.

Q2-2. Since all the experiments are performed in vitro, there is no information on the toxicity of GDS-23. A murine model with/without treatment with GDS-23 can address that. It is also important to know if there are off target effects of the PEG lipid.

A2-2. Currently, we are examining the effects on 3D tissue culture and obtaining results were similar to those in 2D culture. We would like to present these results in a subsequent paper. Furthermore, as off-target effects are important, we plan to demonstrate global effects through comprehensive analyses such as RNA sequencing in the coming paper.

Q2-3. Apart from Nrf-2, what other factors are involved on the GDS-23 treatment? RNAseq of cells treated with/without GDS-23 can be performed to understand the global effect of the lipid.

A2-3. Other aspects involving treatment with GDS-23 have not been clarified yet, and we would like to investigate them in the future.

Q2-4. Antioxidant effect of Nrf-2 is known to influence mitochondrial functions. Did the authors check if there is the effect of GDS-23 on mitochondria?

A2-4. As for the impact on mitochondria, we have not examined it yet, but we would like to do so in the future.

---

## [Decision Letter · Decision Letter 1]

29 Aug 2023

PONE-D-23-15747R1Noncanonical mechanism of Nrf2 activation by diacylglycerol polyethylene glycol adducts in normal human epidermal keratinocytesPLOS ONE

Dear Dr. Miyoshi,

Thank you for submitting your manuscript to PLOS ONE. After careful consideration, we feel that it has merit but does not fully meet PLOS ONE’s publication criteria as it currently stands. Therefore, we invite you to submit a revised version of the manuscript that addresses the points raised during the review process.

We look forward to receiving your revised manuscript.

Kind regards,

Arunava Roy, Ph.D.

Academic Editor

PLOS ONE

Journal Requirements:

Reviewers' comments:

Reviewer's Responses to Questions

**Comments to the Author**

1. If the authors have adequately addressed your comments raised in a previous round of review and you feel that this manuscript is now acceptable for publication, you may indicate that here to bypass the “Comments to the Author” section, enter your conflict of interest statement in the “Confidential to Editor” section, and submit your "Accept" recommendation.

Reviewer #1: All comments have been addressed

Reviewer #2: All comments have been addressed

2. Is the manuscript technically sound, and do the data support the conclusions?

Reviewer #1: Yes

Reviewer #2: Yes

3. Has the statistical analysis been performed appropriately and rigorously? 

Reviewer #1: N/A

Reviewer #2: Yes

4. Have the authors made all data underlying the findings in their manuscript fully available?

Reviewer #1: Yes

Reviewer #2: Yes

5. Is the manuscript presented in an intelligible fashion and written in standard English?

Reviewer #1: Yes

Reviewer #2: Yes

6. Review Comments to the Author

Reviewer #1: 1. The Nrf2 Nuclear Translocation Data is not experimentally convincing without quantification or negative control. However, the supporting data of upstream and downstream mediator activation justifies the involvement of Nrf2 in the process.

2. The effect of GDS-23 on other skin cells or in vivo would hold more significance but the data in the current paper justifies for the study of direct impact of GDS-23 on Keratinocytes.

3. The authors have answered all questions and incorporated supporting data or explanation as applicable.

Hence, I recommend acceptance of the Manuscript for Publication.

Reviewer #2: Authors have addressed most of my comments sufficiently. While I'd still suggest structural control to be necessary for specificity, considering it has been addressed as study drawback is satisfactory.

Since there is no in vivo data, authors may include the idea of using 3D/2D culture in the discussion as future directions for the study.

7. PLOS authors have the option to publish the peer review history of their article (what does this mean?). If published, this will include your full peer review and any attached files.

Reviewer #1: No

Reviewer #2: No

---

## [Author Response · Author response to Decision Letter 1]

5 Sep 2023

Response to Reviewers

Dear Editor

I am writing to inform you of the corrections made to the reference list of our manuscript in accordance with the journal's requirements.

1. We identified an error in the patent number of one of our references and have made the necessary correction.

2. Regarding reference number 2, the absence of a DOI number made it unclear whether the paper had been retracted. To avoid any ambiguity, we have replaced it with another paper from the same author group.

3. Additionally, we noticed that the display format of the DOI numbers in our references was not consistent. We have now standardized it for clarity.

Please review the updated reference list and let us know if there are any further issues. We appreciate your guidance and patience throughout this process.

Dear Reviewer #2

Thank you for your constructive feedback on our manuscript. In response to Reviewer #2's suggestions, we have made the following revisions: As recommended, we have added a section in the discussion addressing the potential of using 3D tissue culture as a future direction for our study. (page 24, lines 416 - 417) We hope these revisions address the reviewer's concerns adequately.

Best regards,

Tatsuro Miyoshi

---

## [Editor Report · Decision Letter 2]

11 Sep 2023

Noncanonical mechanism of Nrf2 activation by diacylglycerol polyethylene glycol adducts in normal human epidermal keratinocytes

PONE-D-23-15747R2

Dear Dr. Miyoshi,

We’re pleased to inform you that your manuscript has been judged scientifically suitable for publication and will be formally accepted for publication once it meets all outstanding technical requirements.

Kind regards,

Arunava Roy, Ph.D.

Academic Editor

PLOS ONE
---

## [Editor Report · Acceptance letter]

13 Sep 2023

PONE-D-23-15747R2 

Noncanonical mechanism of Nrf2 activation by diacylglycerol polyethylene glycol adducts in normal human epidermal keratinocytes 

Dear Dr. Miyoshi:

I'm pleased to inform you that your manuscript has been deemed suitable for publication in PLOS ONE. Congratulations! Your manuscript is now with our production department. 

Kind regards, 

on behalf of

Dr. Arunava Roy 

Academic Editor

PLOS ONE